# Body composition patterns among normal glycemic, pre-diabetic, diabetic health Chinese adults in community: NAHSIT 2013–2016

Sheng-Feng Lin[1,2,3], Yen-Chun Fan[1], Chia-Chi Chou[4,5,6], Wen-Harn Pan[7], Chyi-Huey Bai ◉[1,8]*

**1** School of Public Health, College of Public Health, Taipei Medical University, Taipei City, Taiwan, **2** Division of Hospitalist, Department of Internal Medicine, Far Eastern Memorial Hospital, New Taipei, Taiwan, **3** Department of Neurology, Far Eastern Memorial Hospital, New Taipei, Taiwan, **4** Institute of Epidemiology and Preventive Medicine, National Taiwan University, Taipei, Taiwan, **5** School of Medicine, Chang Gung University, Taoyuan, Taiwan, **6** Department of Internal Medicine, Chang Gung Memorial Hospital, Keelung, Taiwan, **7** Institute of Biomedical Sciences, Academia Sinica, Taipei City, Taiwan, **8** Department of Public Health, College of Medicine, Taipei Medical University, Taipei, Taiwan

* baich@tmu.edu.tw

**Data Availability Statement:** Although a de-identified data set was used, data cannot be shared publicly because of legal restrictions imposed by the government of Taiwan on the distribution of the

## Abstract

### Background

Central obesity is known to be associated with diabetes. Increasing lower extremity circumference was hypothesized in association with lower risk of diabetes.

### Objective

This study determined which anthropometric patterns correlates the best with pre-diabetic and diabetic status among healthy adults.

### Design

Cross-sectional study with nationwide population sampling of participants was designed.

### Participants

In total, 1,358 ethnic Chinese adult participants were recruited from the Nutrition and Health Survey in Taiwan 2013–2016; the whole-body composition was measured through dual-energy X-ray absorptiometry.

### Main outcome measures

Fat and lean mass in whole and specific parts of body among heathy Asian adults with normal glycemic, pre-diabetic, and diabetic states were measured, separately.

### Statistical analyses performed

The generalized linear model was used to investigate the association between body composition (lean and fat mass) and hyperglycemic status. The reduced rank regression (RRR)

personal health data in relation to the "Personal Information Protection Act." Data are only available from the formal proposal to the Health and Welfare Data Science Center (HWDC), Ministry of Health and Welfare, Taiwan for researchers who meet the criteria for access to confidential data. The contact information of Ministry of Health and Welfare was as follows: Address: No.488, Sec. 6, Zhongxiao E. Rd., Nangang Dist., Taipei City 115204, Taiwan (R. O.C.); Tel: (+886)2-8590-6666; Fax: (+886)2-8590-6000.

**Funding:** This work was funded by the Health Promotion Administration, Ministry of Health and Welfare (MOHW108-HPA-H-114-134703, MOHW109-HPA-H-114-144702), as well as funded by Ministry of Science and Technology, Taiwan in the form of a grant awarded to CHB (MOST 107-2314-B-038-072-MY3). The Health Promotion Administration, Ministry of Health and Welfare, Taiwan is the governmental entity. The funders had no role in study design, data collection and analysis, decision to publish, or preparation of the manuscript.

**Competing interests:** The authors have declared that no competing interests exist.

was used to confirm the correlation between glycemic status and predicting factors (body composition parameters).

## Results

Trunk fat positively correlated with the fasting glucose level ($r$ = 0.327, $P$ < 0.001) and HbA1c ($r$ = 0.329, $P$ < 0.001), whereas limb fat negatively correlated with the fasting glucose level ($r$ = −0.325, $P$ < 0.001) and HbA1c ($\rho$ = −0.342, $P$ < 0.001), respectively. In RRR analyses, fasting glucose and HbA1c exhibited a high positive association on fat amount per lean mass of the trunk (factor loading = 0.5319 and 0.5599, respectively) and of android area (0.6422 and 0.6104) and a high negative association fat amount per lean mass of the legs (−0.3863 and −0.3083) and gynoid area (−0.3414 and −0.3725).

## Conclusions

For healthy community participants, increasing trunk fat had a greater risk of hyperglycemic status. Increasing lower extremity mass may confer lower risk of diabetes.

## Introduction

Central obesity is markedly associated with numerous hazardous health effects, such as cardiovascular diseases [1, 2], increased insulin resistance [3, 4], and diabetes mellitus [3, 5]. The prevalence of obesity has increased rapidly in Taiwan [6, 7] and other Asia-Pacific regions [8]. Studies [9, 10] have indicated that changes in dietary habits and lifestyles among residents of Taiwan, such as increasing consumption of sweetened beverages, decreasing physical activity, and increasing sedentary lifestyle, are contributing to the increasing prevalence of metabolic syndrome and central obesity. The Nutrition and Health Survey in Taiwan (NAHSIT), a nationwide research program, assesses the nutritional status and the association between dietary patterns and health among dwellers in Taiwan [6, 11]. Additionally, three waves of the NAHSIT from 1993–1996, 2005–2008, to 2013–2016 have indicated a 2-fold increase in the prevalence of obesity (BMI $\geq$ 27 kg/m$^2$) from 11.8% to 22.0% [6] and diabetes mellitus from 5.3% to 9.1% or above [12] in Taiwan.

Numerous single measurement of the adiposity indices, such as BMI, waist circumference, and waist–hip ratio, are simple surrogate markers for central obesity, but they are limited by overlooking the lean and fat mass distribution in adults [13–15]. Markers of measurement of visceral fat only also limited by neglecting the protective effects of increasing lean mass in hips and lower limbs [16, 17]. Dual-energy X-ray absorptiometry (DXA) is accurate in measuring fat and lean mass in whole and each specific parts of body [18]. We sought to determine the anthropometric patterns which correlates the best for pre-diabetic and diabetic healthy adults.

## Materials and methods

### Subjects

In the NAHSIT 2013–2016 study, a team of well-trained interviewers and technicians conducted a nationwide cross-sectional assessment of the nutritional status and the association between dietary patterns and health among the general population of Taiwan (ethnic Chinese) during 2013–2016. The nutritional evaluation was continued and extending from previous

NAHSIT 2005–2008 study [19], including both the behavior and health outcome indicators, such as questionnaires of dietary recall, of food frequency and habits, of dietary and nutritional knowledge, and serum measurement for clinical biochemistry analytes. The detailed items were described elsewhere [19]. By using a three-stage probability sampling covering 359 townships or city districts, the representative participants for the general Taiwanese population were selected [20]. Additionally, this team obtained health information from the participants by door-to-door visits, and using a standardized nutritional questionnaire. Individual socio-economic status (iSES) was assessed by a sum of z scores of personal income and education years. For iSES, the lowest, middle, and highest classes corresponded to tertiles of total z scores of personal income and education years. The participants' serum specimens were tested by a centralized laboratory. Finally, from the NAHSIT 2013–2016 study, 1,358 participants who completed the body composition examination and laboratory tests of fasting glucose (FG) and HbA1c were included in the present study. This study was approved by the Institutional Review Board on Biomedical Science Research, Academia Sinica, Taiwan (AS-IRB02-103003) and Research Ethics Committee, National Health Research Institutes, Taiwan (EC1020110). Data cannot be shared publicly and uploaded because of legal restrictions imposed by the government of Taiwan on the distribution of the personal health data in relation to the "Personal Information Protection Act." Data are only available from the formal proposal to the Health and Welfare Data Science Center (HWDC), Ministry of Health and Welfare, Taiwan for researchers who meet the criteria for access to confidential data.

## Definition of prediabetes and diabetes

Blood samples were collected for measuring FG and HbA1c. Definition of prediabetes and diabetes is according to the latest standards of American Diabetes Association [21]. For FG, levels of $<100$ mg/dl, 100–125 mg/dl, and $>125$ mg/dl were categorized into normal, prediabetic, and diabetic groups, respectively [21]. HbA1c levels of $<5.7\%$, 5.7–6.4%, and $>6.4\%$ were classified into normal, prediabetic, and diabetic groups, respectively [21]. Diagnosis of diabetes mellitus was confirmed on the basis of self-reported diabetes with treatment, or laboratory tests with FG $\geq 126$ mg/dL, or HbA1c $\geq 6.5\%$.

## Data collection of body composition

Demographic information, and results of anthropometric assessment of weight, waist circumference, and BMI were recorded. Body composition was measured using the mobile DXA (Prodigy, GE Lunar Health Care, Wisconsin, USA) when conducting the nutritional status survey. Furthermore, the body composition parameters were defined as follows:

- Fat body weight (%): (Total fat mass/1000)/ Body weight $\times$ 100

- Limb fat body weight (%): ((Arms fat mass + Legs fat mass)/1000)/ Body weight $\times$ 100

- Trunk fat body weight (%): (Trunk fat mass)/1000)/ Body weight $\times$ 100

- Lean body weight (%): (Total lean mass/1000)/ Body weight $\times$ 100

- Limb lean body weight (%): ((Arms lean mass + Legs lean mass)/1000)/ Body weight $\times$ 100

- Trunk lean body weight (%): (Trunk lean mass)/1000)/ Body weight $\times$ 100

- Limb in fat (%): (Arms fat mass + Legs fat mass)/Total fat mass $\times$ 100

- Arms in fat (%): (Arms fat mass/Total fat mass) $\times$ 100

- Legs in fat (%): (Legs fat mass/Total fat mass) $\times$ 100

- Android in fat (%): (Android fat mass/Total fat mass) × 100

- Gynoid in fat (%): (Gynoid fat mass/Total fat mass) × 100

- Total region fat (%): (Fat mass in limbs and trunk/ Body weight) × 100

- Total tissue fat (%): (Total fat mass/ Body weight) × 100

## Statistical analysis

Continuous variables were expressed as mean and standard deviation and analyzed using the Kruskal–Wallis H test. The Pearson correlation statistics was used to examine the association between the blood glucose or HbA1c level and body composition markers. The generalized linear model (GLM) was used to estimate body composition parameters according to the blood glucose and HbA1c levels after adjustment for covariates of age, sex, blood pressure, and high density lipoprotein cholesterol (HDL-C). Reduced rank regression (RRR) is a modern method in epidemiology assessing the role of the response variables and latent variables [22, 23]. By using RRR, body composition patterns associated with blood glucose and HbA1c were derived. Sensitivity analysis was performed by excluding participants with diabetes mellitus who received treatment. All analyses were performed using SAS software 9.4 (Cary, NC). An alpha level of <0.05 was defined as statistically significant.

## Sensitivity analyses

Two sensitivity analyses were performed. First, we analyzed the association between body composition markers and different blood glucose and HbA1c levels by excluding patients of diabetes mellitus with medical treatment. Second, we investigated the relationship between body composition markers and insulin resistance (IR) by triglyceride glucose-waist circumference index (TyG-WC index) [24, 25]. The TyG and TyG-related markers have been used as the surrogate indicators for IR [25, 26]. Of these markers, the TyG-WC index was more accurate in estimating IR [25]. The formula of the TyG-WC index is Ln (triglyceride [mg/dL] × fasting glucose [mg/dL]/2) × waist circumference [24, 25]. A cutoff value of TyG-WC index $\geq$ 850 indicates IR [25, 26].

## Results

### Participant characteristics

Among the enrolled 1,358 participants, 214 (15.8%) had a confirmed diagnosis of diabetes mellitus (Table 1). The average FG and HbA1c levels in the diabetic and nondiabetic groups were 144.9 mg/dL and 97.0 mg/dL, and 7.3% and 5.6%, respectively. The proportions of male participants, levels of blood pressure and triglycerides, and the ever and current smokers were higher in the diabetic group. The average HDL-C level and z score of iSES were lower in the diabetic group. The nondiabetic group had higher SES when compared to the diabetic group. Between the diabetic and nondiabetic groups, alcohol consumption and physical activity was no significantly difference. Besides, 50.9% (109/214) of the diabetic participants took anti-diabetic medications.

### Body composition markers on fasting glucose level

Participants were categorized into three groups based on the FG levels of $\geq$126 mg/dL, 100–125 mg/dL, and <100 mg/dL (Table 2). Group with FG < 100 mg/dL exhibited significantly

**Table 1. Markers that related obesity to diabetes mellitus (n = 1358).**

| Markers, units | Total | | Diabetes mellitus | | | | P value[a] |
|---|---|---|---|---|---|---|---|
| | n = 1358 | | Yes (n = 214) | | No (n = 1144) | | |
| | Mean | SD | mean | SD | mean | SD | |
| Age, years | 52.2 | 16.9 | 62.4 | 11.8 | 50.4 | 17.0 | <0.001* |
| Male, n (ratio) | n = 688 | (50.7%) | n = 128 | (59.8%) | n = 560 | (49.0%) | 0.004* |
| HbA1c, % | 5.9 | 0.9 | 7.3 | 1.4 | 5.6 | 0.4 | <0.001* |
| Glucose, mg/dL | 104.6 | 26.9 | 144.9 | 47.5 | 97.0 | 8.8 | <0.001* |
| SBP[b], mmHg | 123.3 | 18.2 | 133.1 | 16.8 | 121.4 | 17.8 | <0.001* |
| DBP[b], mmHg | 75.2 | 10.6 | 77.1 | 11.1 | 74.9 | 10.5 | 0.008* |
| TG, mg/dL | 117.3 | 79.4 | 163.3 | 106.2 | 108.7 | 70.1 | <0.001* |
| HDL, mg/dL | 54.3 | 15.6 | 47.2 | 13.2 | 55.7 | 15.7 | <0.001* |
| Smoking[c] | | | | | | | |
| Non-smokers, n (ratio) | n = 905 | (69.7%) | n = 133 | (63.0%) | n = 772 | (71.0%) | 0.072 |
| Ever smokers, n (ratio) | n = 222 | (17.1%) | n = 44 | (20.9%) | n = 178 | (16.4%) | |
| smokers, n (ratio) | n = 172 | (13.2%) | n = 34 | (16.1%) | n = 138 | (12.7%) | |
| Alcohol consumption[d] (g/d) | 21.2 | 100.6 | 32.1 | 135.7 | 19.0 | 91.8 | 0.838 |
| Non-drinkers, n (ratio) | n = 582 | (56.0%) | n = 103 | (58.2%) | n = 479 | (55.6%) | 0.123 |
| < 16 g/d, n (ratio) | n = 273 | (26.3%) | n = 36 | (20.3%) | n = 237 | (27.5%) | |
| 16–35 g/d, n (ratio) | n = 72 | (6.9%) | n = 11 | (6.2%) | n = 61 | (7.1%) | |
| 36–64 g/d, n (ratio) | n = 35 | (3.4%) | n = 9 | (5.1%) | n = 26 | (3.0%) | |
| > 64 g/d, n (ratio) | n = 77 | (7.4%) | n = 18 | (10.2%) | n = 59 | (6.8%) | |
| iSES level (z score) | 0.0 | 1.6 | -0.53 | 1.6 | 0.10 | 1.6 | <0.001* |
| Lowest SES, n (ratio) | n = 446 | (32.8%) | n = 97 | (45.3%) | n = 349 | (30.5%) | <0.001* |
| Middle SES, n (ratio) | n = 459 | (33.8%) | n = 69 | (32.2%) | n = 390 | (34.1%) | |
| Highest SES, n (ratio) | n = 453 | (33.4%) | n = 48 | (22.4%) | n = 405 | (35.4%) | |
| Physical activity[e], MET hours/week | 5.3 | 24.2 | 5.3 | 22.8 | 5.3 | 24.5 | 0.570 |

Data are expressed as mean and standard deviation (SD) for continuous variables and number (%) for discrete variables. DBP, diastolic blood pressure; HDL, high-density lipoprotein; iSES, individual socioeconomic status; MET, metabolic equivalent of task; n, number; SBP, systolic blood pressure; TG, triglycerides.

*Statistical significance as $P < 0.05$.

[a]Tested by the Mann-Whitney U test and Chi-square test.

[b]n = 1277.

[c]n = 1299.

[d]n = 1039.

[e]n = 1045.

lower weight, BMI, waist circumference, but heavier limb weight in both fat and lean mass than the groups with higher FG levels. Three groups showed no significant difference for lean body weight in total body and trunk area. The Pearson correlation statistics revealed modest positive correlation between FG and tissue percentage of trunk in fat mass ($r = 0.327$, $P <0.001$), negative correlation between FG and limb in fat mass percentage ($r = -0.325$, $P <0.001$) (Table 3). With increasing of FG levels, GLM regression analyses revealed a consistent increase of trunk weight and decrease of limb weight in both fat and lean mass.

## Body composition markers on HbA1c level

Based on the HbA1c levels, participants were categorized into three groups of ≥6.5%, 5.7–6.4%, and <5.7% (Table 4). Similar to the results for FG levels, groups with lower HbA1c levels

**Table 2. Body composition markers that related obesity to fasting glucose levels (n = 1358).**

| Markers, unit | Fasting Glucose (mg/dL) | | | | | | P value[a] |
|---|---|---|---|---|---|---|---|
| | ≥126 (n = 139) | | 100–125 (n = 456) | | <100 (n = 763) | | |
| | Mean | SD | Mean | SD | Mean | SD | |
| Weight, kg | 68.7 | 13.4 | 67.9 | 12.4 | 62.0 | 12.6 | <0.001* |
| BMI, kg/m² | 26.3 | 4.2 | 25.5 | 3.6 | 23.5 | 3.8 | <0.001* |
| Waist, cm | 91.6 | 10.4 | 88.2 | 10.4 | 80.8 | 10.3 | <0.001* |
| Total fat mass, g | 22070.3 | 7830.1 | 21249.5 | 7023.3 | 18651.4 | 7009.0 | <0.001* |
| Total lean mass, g | 43235.6 | 8362.4 | 43121 | 8812.0 | 40009.3 | 8652.7 | <0.001* |
| Total region fat, % | 32.1 | 7.8 | 31.6 | 7.9 | 30.3 | 8.6 | 0.026* |
| Total tissue fat, % | 33.3 | 8.0 | 32.8 | 8.1 | 31.5 | 8.8 | 0.035* |
| Fat body weight, % | 31.7 | 7.7 | 31.2 | 7.9 | 29.8 | 8.6 | 0.024* |
| Limb fat body weight, % | 11.6 | 3.7 | 12.1 | 3.9 | 12.9 | 4.3 | <0.001* |
| Trunk fat body weight, % | 19.0 | 4.7 | 18.0 | 4.5 | 15.8 | 4.9 | <0.001* |
| Lean body weight, % | 63.3 | 7.3 | 63.7 | 7.6 | 64.8 | 8.4 | 0.078 |
| Limb lean body weight, % | 27.3 | 3.5 | 28.2 | 4.0 | 28.5 | 4.2 | 0.018* |
| Trunk lean body weight, % | 31.1 | 4.0 | 30.4 | 3.8 | 31.0 | 4.2 | 0.067 |
| Limb in fat, % | 36.2 | 5.3 | 38.4 | 5.6 | 43.0 | 6.0 | <0.001* |
| Trunk in fat, % | 60.0 | 5.4 | 57.9 | 5.7 | 52.8 | 6.4 | <0.001* |
| Limb in lean, % | 43.2 | 2.6 | 44.3 | 2.6 | 44.0 | 2.5 | <0.001* |
| Trunk in lean, % | 49.1 | 2.3 | 47.8 | 2.1 | 47.8 | 2.0 | <0.001* |

BMI, body mass index.

*Statistical significance as P < 0.05.

[a]Kruskal–Wallis H test

**Table 3. Body composition markers that related obesity to blood glucose levels (n = 1358).**

| Markers, unit | Pearson correlation (glucose) | P value | Regression coefficient (glucose)[a] | | P value |
|---|---|---|---|---|---|
| | | | Beta | 95% CI | |
| Weight, kg | 0.143 | <0.001 | 0.03496 | (0.0109 to 0.05902) | 0.004* |
| BMI, kg/m² | 0.184 | <0.001 | 0.01184 | (0.00366 to 0.02001) | 0.005* |
| Waist, cm | 0.268 | <0.001 | 0.04082 | (0.0197 to 0.06193) | <0.001* |
| Total fat mass, g | 0.109 | <0.001 | 15.5139 | (-2.3243 to 33.3521) | 0.088 |
| Total lean mass, g | 0.121 | <0.001 | 21.27296 | (10.17281 to 32.3731) | <0.001* |
| Total region fat, % | 0.035 | 0.204 | 0.00194 | (-0.01467 to 0.01855) | 0.819 |
| Total tissue fat, % | 0.032 | 0.241 | 0.00197 | (-0.01511 to 0.01905) | 0.821 |
| Fat body weight, % | 0.036 | 0.188 | 0.00002 | (-0.00015 to 0.00019) | 0.802 |
| Limb fat body weight, % | -0.119 | <0.001 | -0.00007 | (-0.00013 to -0.00001) | 0.019* |
| Trunk fat body weight, % | 0.164 | <0.001 | 0.00012 | (0.00001 to 0.00024) | 0.040* |
| Lean body weight, % | -0.022 | 0.416 | 0.00002 | (-0.00012 to 0.00015) | 0.798 |
| Limb lean body weight, % | -0.059 | 0.030 | -0.00004 | (-0.00009 to 0.00002) | 0.183 |
| Trunk lean body weight, % | 0.037 | 0.177 | 0.00007 | (-0.00001 to 0.00014) | 0.090 |
| Limb in fat, % | -0.325 | <0.001 | -0.00026 | (-0.00034 to -0.00018) | <0.001* |
| Trunk in fat, % | 0.327 | <0.001 | 0.00035 | (0.00023 to 0.00046) | <0.001* |
| Limb in lean, % | -0.097 | <0.001 | -0.00007 | (-0.00012 to -0.00003) | 0.002* |
| Trunk in lean, % | 0.173 | <0.001 | 0.00009 | (0.00005 to 0.00013) | <0.001* |

[a]Tested by GLM and adjusted age, sex, systolic blood pressure, diastolic blood pressure, triglycerides, HDL, and iSES level. (n = 1277).

Table 4. Body composition markers that related obesity to HbA1c levels (n = 1358).

| Markers, units | HbA1c (%) | | | | | | P value[a] |
| --- | --- | --- | --- | --- | --- | --- | --- |
| | ≥6.5 (n = 177) | | 5.7–6.4 (n = 530) | | <5.7 (n = 651) | | |
| | Mean | SD | mean | SD | Mean | SD | |
| Weight, kg | 68.8 | 13.6 | 65.2 | 12.6 | 63.15 | 12.79 | <0.001* |
| BMI, kg/m² | 26.3 | 4.1 | 24.9 | 3.8 | 23.63 | 3.82 | <0.001* |
| Waist, cm | 91.5 | 10.3 | 86.0 | 10.8 | 81.14 | 10.46 | <0.001* |
| Total fat mass, g | 22055.8 | 7556.8 | 20274.6 | 7194.3 | 18954.11 | 7027.83 | <0.001* |
| Total lean mass, g | 43231.7 | 8498.8 | 41433.3 | 8912.2 | 40842.31 | 8753.02 | 0.003* |
| Total region fat, % | 32.2 | 7.6 | 31.4 | 8.5 | 30.20 | 8.32 | 0.003* |
| Total tissue fat, % | 33.4 | 7.8 | 32.6 | 8.7 | 31.42 | 8.56 | 0.006* |
| Fat body weight, % | 31.7 | 7.6 | 30.9 | 8.4 | 29.84 | 8.36 | 0.01* |
| Limb fat body weight, % | 11.6 | 3.7 | 12.2 | 4.1 | 12.91 | 4.22 | <0.001* |
| Trunk fat body weight, % | 18.9 | 4.4 | 17.5 | 4.9 | 15.79 | 4.79 | <0.001* |
| Lean body weight, % | 63.2 | 7.2 | 63.8 | 8.1 | 64.96 | 8.13 | 0.004* |
| Limb lean body weight, % | 27.4 | 3.4 | 28.1 | 4.0 | 28.75 | 4.18 | <0.001* |
| Trunk lean body weight, % | 30.8 | 4.0 | 30.6 | 4.1 | 30.92 | 4.05 | 0.177* |
| Limb in fat, % | 36.3 | 5.2 | 39.4 | 5.7 | 43.13 | 6.20 | <0.001* |
| Trunk in fat, % | 59.9 | 5.3 | 56.6 | 6.0 | 52.82 | 6.58 | <0.001* |
| Limb in lean, % | 43.4 | 2.7 | 44.0 | 2.5 | 44.22 | 2.62 | 0.008* |
| Trunk in lean, % | 48.7 | 2.4 | 48.0 | 2.0 | 47.61 | 2.10 | <0.001* |

BMI, body mass index.

*Statistical significance as P < 0.05.

[a]Kruskal–Wallis H test

exhibited greater decrease in weight, BMI, waist circumference, total fat mass, total lean mass, total region fat, and tissue fat. Limb in fat showed moderate negative ($r$ = -0.342, $P$ <0.001) and trunk in fat revealed positive correlations with HbA1c ($r$ = 0.329, $P$ <0.001), respectively (Table 5). With increasing levels of HbA1c, our regression analyses consistently revealed decrease of limb weight and increase of trunk weight for both lean and fat mass.

## Sensitivity analyses

First, on excluding patients of diabetes mellitus with medical treatment, the effects of body composition markers on different blood glucose and HbA1c levels were found coherent to our primary analyses, as mentioned previously (S1–S3 Tables). Second, 236 and 1122 participants were respectively categorized into with IR and non-IR groups (S4 Table). Consistent to the participants with higher FG or HbA1c, the IR group showed increased values of weight, BMI, waist, total fat mass, and markers of limb in fat (%), and trunk in fat (%). With increasing values of TyG-WC index, the participants showed decrease of lean mass weight and increase of fat mass weight (S5 Table).

## Body composition parameter patterns explained by fasting glucose and HbA1c

RRR analyses revealed that FG and HbA1c shared similar factor loadings (Table 6). Both FG and HbA1c exhibited higher positive factor loadings (explaining the positive correlation between the observed variables of body composition and latent variable of FG or HbA1c) for

Table 5. Body composition markers that related obesity to HbA1c levels (n = 1358).

| Markers, units | Pearson correlation (HbA1c) | P value | Regression coefficient (HbA1c)[a] | | P value |
|---|---|---|---|---|---|
| | | | Beta | 95% CI | |
| Weight, kg | 0.096 | <0.001 | 0.84539 | (0.12748 to 1.5633) | 0.021* |
| BMI, kg/m$^2$ | 0.165 | <0.001 | 0.33533 | (0.09016 to 0.58051) | 0.007* |
| Waist, cm | 0.242 | <0.001 | 0.96860 | (0.33674 to 1.60046) | 0.003* |
| Total fat mass, g | 0.082 | 0.003 | 196.49335 | (-315.76508 to 708.75177) | 0.452 |
| Total lean mass, g | 0.073 | 0.007 | 615.02581 | (281.70332 to 948.3483) | <0.001* |
| Total region fat, % | 0.036 | 0.180 | -0.20021 | (-0.69218 to 0.29177) | 0.425 |
| Total tissue fat, % | 0.033 | 0.225 | -0.21164 | (-0.71762 to 0.29434) | 0.412 |
| Fat body weight, % | 0.033 | 0.220 | -0.00230 | (-0.00728 to 0.00268) | 0.366 |
| Limb fat body Weight, % | -0.128 | <0.001 | -0.00363 | (-0.00548 to -0.00178) | <0.001* |
| Trunk fat body weight, % | 0.164 | <0.001 | 0.00208 | (-0.00129 to 0.00546) | 0.226 |
| Lean body weight, % | -0.029 | 0.279 | 0.00166 | (-0.00244 to 0.00576) | 0.427 |
| Limb lean body weight, % | -0.084 | 0.002 | -0.00047 | (-0.00223 to 0.00129) | 0.599 |
| Trunk lean body weight, % | 0.043 | 0.113 | 0.00239 | (0.00007 to 0.00471) | 0.043* |
| Limb in fat, % | -0.342 | <0.001 | -0.00902 | (-0.01155 to -0.00648) | <0.001* |
| Trunk in fat, % | 0.329 | <0.001 | 0.01088 | (0.00745 to 0.0143) | <0.001* |
| Limb in lean, % | -0.137 | <0.001 | -0.00189 | (-0.0033 to -0.00048) | 0.009* |
| Trunk in lean, % | 0.206 | <0.001 | 0.00250 | (0.00123 to 0.00378) | <0.001* |

[a] Tested by GLM and adjusted age, sex, systolic blood pressure, diastolic blood pressure, triglycerides, HDL, and iSES level. (n = 1277)

the trunk in lean percentage (0.2983 and 0.3093, respectively) and the trunk in fat percentage (0.4282 and 0.3909, respectively) and higher negative loadings for the legs in lean percentage (−0.2595 and −0.2864, respectively) and the legs in fat percentage (−0.4514 and −0.4545, respectively). The total lean mass and total fat mass in grams also showed a similar pattern.

On stratification by lean mass, FG and HbA1c still exhibited the highest positive loadings on the android area [(0.7008 and 0.7727 in grams, respectively) and (0.6216 and 0.6285 in percentage, respectively)] and high negative loadings on the legs (−0.4636 and −0.4858 in percentage, respectively) and the gynoid area (−0.4575 and −0.2908 in percentage, respectively). On stratification by fat tissue, FG and HbA1c still exhibited the highest positive loading on the android area [(0.4762 and 0.4285 in percentage, respectively) and (0.7019 and 0.6497 in grams, respectively)] and a high negative loading on the legs (−0.5447 and −0.5627 in percentage, respectively) and the gynoid area (−0.4575 and −0.4958 in percentage, respectively). For the index of fat amount per lean mass ratio (fat mass/lean mass), FG and HbA1c shared high positive loadings on the trunk (0.5319 and 0.5599, respectively) and the android area (0.6422 and 0.6104, respectively) and negative loadings on the legs (−0.3863 and −0.3083, respectively) and the gynoid area (−0.3414 and −0.3725, respectively).

## Discussion

This study offered comprehensive exploration for the association between body composition parameters, measured through DXA, and normal glycemic, prediabetic, and diabetic status, defined by FG and HbA1c levels, with a nationwide sampling ethnic Chinese population. On analyzing the body composition, we found that trunk fat positively correlated with increasing FG and HbA1c levels. On the contrary, lower extremity weight (in both fat and lean mass) showed negatively correlation with FG and HbA1c levels.

**Table 6. RRR loading of blood glucose and HbA1c levels.**

| Body Area Loading | Unit: percentage (%) | | Unit: gram | |
|---|---|---|---|---|
| | Glucose (mg/dL) | HbA1c (%) | Glucose (mg/dL) | HbA1c (%) |
| Total lean and fat mass | (a) Total body composition (%) | | (b) Total body composition | |
| Arms in lean | 0.0583 | 0.2030* | 0.1375 | 0.0564 |
| Legs in lean | −0.2595* | 0.0915 | −0.0266 | −0.2864* |
| Trunk in lean | 0.2983* | 0.4056* | 0.3274* | 0.3093* |
| Android in lean | 0.3479* | 0.4742*† | 0.4363*† | 0.3705* |
| Gynoid in lean | −0.1803 | 0.1374 | 0.0416 | −0.1715 |
| Arms in fat | 0.0130 | 0.2089* | 0.3165* | 0.1135 |
| Legs in fat | −0.4514*† | −0.1476 | −0.2006* | −0.4545*† |
| Trunk in fat | 0.4282*† | 0.4478*† | 0.4740*† | 0.3909* |
| Android in fat | 0.3946* | 0.5168*† | 0.5362*† | 0.3461* |
| Gynoid in fat | −0.3791* | −0.0952 | −0.1686 | −0.4004*† |
| Stratification by lean mass | (c) Total lean mass (%) | | (d) Total lean mass (g) | |
| Arms | 0.1041 | 0.3001* | 0.2435* | 0.0957 |
| Legs | −0.4636*† | 0.1352 | −0.0470 | −0.4858*† |
| Trunk | 0.5330*† | 0.5994*† | 0.5797*† | 0.5247*† |
| Android | 0.6216*† | 0.7008*† | 0.7727*† | 0.6285*† |
| Gynoid | −0.4575*† | 0.2031* | 0.0737 | −0.2908* |
| Stratification by fat mass | (e) Total fat mass (%) | | (f) Total fat mass (g) | |
| Arms | 0.0157 | 0.2837* | 0.3835* | 0.1406 |
| Legs | −0.5447*† | −0.2004* | −0.2431* | −0.5627*† |
| Trunk | 0.5167*† | 0.6082*† | 0.5744*† | 0.4840*† |
| Android | 0.4762*† | 0.7019*† | 0.6497*† | 0.4285*† |
| Gynoid | −0.4575*† | −0.1293 | −0.2043* | −0.4958*† |
| | (g) Fat amount per lean (Ratio of Fat mass/Lean mass) | | | |
| Arms | 0.1290 | 0.3198* | | |
| Legs | −0.3863* | −0.3083* | | |
| Trunk | 0.5319*† | 0.5599*† | | |
| Android | 0.6422*† | 0.6104*† | | |
| Gynoid | −0.3725* | −0.3414* | | |

*Factor loading > 0.2.

†Factor loading > 0.4.

Our study consistently showed two major patterns between anthropometric parameters and glycemic status: (a) increasing levels of FG and HbA1c increased trunk fat mass and trunk lean mass, and (b) increasing lower extremity weight (either in fat or in lean mass) conferred decreased levels of FG and HbA1c. For the first pattern, this was consistent to the finding that central obesity was more common in type 2 diabetes [3, 5, 8]. Compared to the second pattern, previous studies showed inconsistent results. A study through analyzing the NHANES 1999–2004 found no significant association in limb lean mass and glycemic status on adjusting age, body mass index, ethnicity, smoking, alcohol, and physical activity [27]. In contrast, some studies found that lean mass is significantly negatively associated with insulin resistance among pre-diabetic patients [28, 29]. We considered that this difference reflect the different severity of insulin resistance among individuals from the sampling population. Lean mass in limbs may be an alternative to insulin resistance as a body composition marker. Lastly, our analyses should

also support the evidence supporting the hypothesis that increasing limb mass decreased risk of diabetes. To validate our findings, we performed a confirmatory RRR analysis.

The RRR was similar to both exploratory principal component analysis and factor analysis, it is superior to these traditional dimension reduction methods because it involves determination of linear functions of the predicting factors (body composition parameters) by maximizing the explained variation in the response variables (blood glucose and HbA1c) [23]. Because RRR employs both information of the response variables (FG and HbA1c) and predicting factors (body composition parameters), it represents a posteriori method and thus validates the correlation between the response variables and predicting factors.

Additionally, the RRR analysis showed that individuals with higher FG and HbA1c levels accumulated more fat mass and increased lean mass in the android area, but less fat and lean mass in their lower extremities. We considered these findings were compatible to previous studies indicating functional differences of fat tissue between upper body (android area) and lower body [30–32]. Previous studies found the opposite association of upper and lower body fat with 2-hour post-load glucose level of oral glucose tolerance test [32] and diabetic risk [33, 34]. Recent studies indicated this difference was medicated by different adipokines [31] and site-specific sets of genes between upper and low body fat tissues [30, 35], respectively. On the other hand, the RRR also showed android lean mass was positively associated with FG and HbA1c. We found some studies showed the similar findings in their original data: diabetic participants with significantly higher trunk lean mass in comparison to non-diabetes [34, 36]. To our recognition, this should be cautious interpreted since DXA may severely overestimate trunk and android lean mass [37, 38]. When compared to magnetic resonance imaging (MRI) modality, a study found DXA overestimate 7% and 48% of trunk and android lean mass, respectively [38]. The association of trunk and android lean mass with FG and HbA1c should be reserved to answer with MRI.

The distinctive strengths of this study are as follows: (1) a nationwide population-based study investigating the association between body composition, measured through DXA, and differed glycemic status among healthy individuals in general communities, (2) a comprehensive exploration of the association between lean mass, fat mass, and blood glucose in different body regions, and (3) the validation of the findings through RRR. Besides, the sensitivity analysis helps confirm the major results and explain the mechanism between body composition markers and glycemic change was by insulin resistance.

The major study limitation, which was not addressed, was that a causal relationship between these body composition indexes and blood glucose levels could not be drawn by using the cross-sectional study design. However, the strong association and our sensitivity analysis results, based on the diagnosis of diabetes, were consistent. Thus, these findings suggest a direct correlation between body composition and blood glucose levels. Second, serum insulin was unmeasured in the NAHSIT 2013–2016. Our investigation of association between body composition and IR was replaced by the TyG-WC index. In fact, the recent studies suggested the TyG-WC index is a reliable alternative marker to homeostasis model assessment of insulin resistance (HOMA-IR) [24, 25]. Third, dynapenia or decreased muscle strength may influence glycemia status [39]. However, the strength was not measured and investigated in this study. Lastly, unlike the hospital-based study, the NAHSIT aimed to enroll healthy participants in the community. The unfavorable health outcome indicators, such as diabetic complications, were not explored in this survey.

Besides, numerous regions or countries in east Asia, like Japan, Korea, and China, was mono-ethnicity. In fact, more than 95% of residents in Taiwan were Han Chinese. A subethnic study in Taiwan also found no significant difference in the prevalence of diabetes and impaired glucose tolerance (IGT) between Taiwanese Aborigines and Han Chinese [40]. Therefore, no

ethnic difference for on body composition, phenotype of type 2 diabetes, and glycemic status were explored in this study.

## Conclusions

In conclusion, abdominal or trunk fat is found strongly associated with hyperglycemia for healthy adults in the general community. High fat and low lean mass percentage in abdomen increases the risk of hyperglycemia. Increasing lower extremity mass may confer lower risk of hyperglycemia. Furthermore, studies with physical activity and lifestyle intervention for increasing lower extremity mass could be implemented among the general population in the future.

## Supporting information

**S1 Table. The body composition markers that related obesity according to diabetes mellitus diagnosis.** (n = 1358)–sensitivity analysis.
(DOCX)

**S2 Table. The body composition markers that related obesity according to diabetes mellitus diagnosis.** (n = 1358)–sensitivity analysis.
(DOCX)

**S3 Table. The body composition markers that related obesity according to DM diagnosis.** (n = 1358)–sensitivity analysis.
(DOCX)

**S4 Table. Body composition markers that related obesity to Triglyceride Glucose-Waist Circumference (TyG-WC) index.**
(DOCX)

**S5 Table. Body composition markers that related obesity to Triglyceride Glucose-Waist Circumference (TyG-WC) index.**
(DOCX)

## Acknowledgments

The content of this research may not represent the opinion of the Health Promotion Administration, Ministry of Health and Welfare.

## Author Contributions

**Conceptualization:** Sheng-Feng Lin, Yen-Chun Fan, Chia-Chi Chou, Wen-Harn Pan, Chyi-Huey Bai.

**Data curation:** Chyi-Huey Bai.

**Formal analysis:** Sheng-Feng Lin, Yen-Chun Fan, Chyi-Huey Bai.

**Funding acquisition:** Chyi-Huey Bai.

**Investigation:** Wen-Harn Pan.

**Supervision:** Wen-Harn Pan, Chyi-Huey Bai.

**Writing – original draft:** Sheng-Feng Lin.

**Writing – review & editing:** Sheng-Feng Lin.

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
