## [Decision Letter · Decision Letter 0]

6 Aug 2020

PONE-D-20-13486

Body Composition Patterns among Normal Glycemic, Pre-diabetic, Diabetic Health Chinese Adults in Community: NAHSIT 2013–2016

PLOS ONE

Dear Dr. Bai,

Thank you for submitting your manuscript to PLOS ONE. After careful consideration, we feel that it has merit but does not fully meet PLOS ONE’s publication criteria as it currently stands. Therefore, we invite you to submit a revised version of the manuscript that addresses the points raised during the review process.

We look forward to receiving your revised manuscript.

Kind regards,

Cheng Hu

Academic Editor

PLOS ONE

Journal Requirements:

2. Please refrain from stating p values as 0.000, either report the exact value or employ the format p<0.001.

3.Thank you for stating the following financial disclosure:

 [The funders had no role in study design, data collection and analysis, decision to publish, or preparation of the manuscript.].

4.We note that you have indicated that data from this study are available upon request. PLOS only allows data to be available upon request if there are legal or ethical restrictions on sharing data publicly. For information on unacceptable data access restrictions, please see http://journals.plos.org/plosone/s/data-availability#loc-unacceptable-data-access-restrictions.

5. Please amend your list of authors on the manuscript to ensure that each author is linked to an affiliation. Authors’ affiliations should reflect the institution where the work was done (if authors moved subsequently, you can also list the new affiliation stating “current affiliation:….” as necessary).

6. Please upload a copy of Figure 1 and 2, to which you refer in your text on page 4. If the figure is no longer to be included as part of the submission please remove all reference to it within the text.

Reviewers' comments:

Reviewer's Responses to Questions

**Comments to the Author**

1. Is the manuscript technically sound, and do the data support the conclusions?

Reviewer #1: Yes

Reviewer #2: Partly

2. Has the statistical analysis been performed appropriately and rigorously? 

Reviewer #1: Yes

Reviewer #2: I Don't Know

3. Have the authors made all data underlying the findings in their manuscript fully available?

Reviewer #1: Yes

Reviewer #2: Yes

4. Is the manuscript presented in an intelligible fashion and written in standard English?

Reviewer #1: Yes

Reviewer #2: Yes

5. Review Comments to the Author

Reviewer #1: The manuscript entitled "Body Composition Patterns among Normal Glycemic, Pre-diabetic, Diabetic Health Chinese Adults in Community: NAHSIT 2013–2016" reports the association between body composition parameters and normal glycemic, prediabetic, and diabetic status in Chinese adult population sampled for the nationwide survey. Although the association between body composition parameters and each glycemic status was clearly described, the authors should discuss some additional points to emphasize a rationale of this study.

Specific comments:

1. The authors should discuss the possible confounding factors of body composition and glycemic status. For example, smoking, alcohol and socioeconomic status could be confounding factors, especially in older adults (Osteoporosis and sarcopenia 2018;4:109-113, Lancet Diabetes Endocrinol 2015;3:958-967, Int J epidemiol 2011;40:804-818; J Diabetes Investig. 2020 Mar 29. doi: 10.1111/jdi.13260. etc).

2. The authors should consider the diabetic complications and anti-diabetes drugs to appropriately assess the relationship between body composition and glycemic status in patients with diabetes. Diabetic neuropathy and use of SU and glinides may affect skeletal muscle mass (J Am Med Dir Assoc 2016;17:896-901, J Diabetes Complications 2017;31:1465-1473, etc).

3. The authors should discuss not only muscle mass but also muscle strength as well as exercise capacity. Dynapenia could also influence glycemic status (J epidemiol 2015 ; 25 : 656-662); and the authors should discuss the possible relationship in the manuscript.

4. The authors should discuss possible effects of ethnic differences on body composition and glycemic status. In the previous study, it was suggested that ethnic differences may affect body composition, diet and physical activity (Womens Health(Lond) 2015;11:913-27).

5. Since the phenotype of type 2 diabetes differ from those of other ethnic groups (Lancet Diabetes Endocrinol. 2016 Jan;4(1):2-3; Curr Diab Rep. 2015 Jun;15(6):602; J Diabetes Investig. 2016 Apr;7 Suppl 1(Suppl 1):102-9 etc). The authors should discuss possible etnic differences in their observed results.

Reviewer #2: Major

1. In line 214-215, you are writing “lean mass…”. Therefore, not only comparison between glucose profile and body composition but also the comparison with an insulin level and HOMA-IR and the body composition is necessary.

2. In line 243-244, you are writing “low lean mass percentage in abdomen…”. Results of RRR show that trunk lean and fat mass have a positive association with a glucose profile. Trunk in fat and lean has a positive association with glucose profile, and limb in fat and lean seems to have a negative association with glucose profile. Therefore, the discussion about this difference seems to be necessary.

Minor

1. In line 85, you are writing “…assessment of the nutritional status”. What kind of item did you use for nutritional evaluation, blood nutritional items (albumin, cholinesterase etc.) or nutrition score (MNA, CONUT etc.)?

2. In the section of Data Collection of Body Composition (117-125), the calculating formula is (body composition/1000)/weight *100, isn't it?

3. In line 150, is the classification of the FG levels ≥126mg/dL, 100-126 mg/dL and <100mg/dL? It is necessary to conform them to data of Table2.

4. In line 161, did you mistake “≥6.5%” for “>6.5%”?

5. In line 205, you wrote "…increase trunk fat mass". But not only "trunk fat mass" but also "trunk lean mass" increased.

6. Explanation of total region fat, total tissue fat, android (in fat) and gynoid (in fat) is necessary in the method.

7. In Table 1, I think the data of Male is "number and ratio" not "mean and SD".

8. The unit of weight is “kg” not “cm” (Table2-5).

9. Please revise the data of SD from 10.41 to 10.4 in the item of the waist in Table 2.

10. Please revise “Statistical significance as P<0.5” to “Statistical significance as P<0.05” in Table 2.

11. I think that HbA1c is unnecessary for a title of Table3.

6. PLOS authors have the option to publish the peer review history of their article (what does this mean?). If published, this will include your full peer review and any attached files.

Reviewer #1: No

Reviewer #2: No

---

## [Author Response · Author response to Decision Letter 0]

7 Sep 2020

We thank the reviewers for their constructive comments. We have made revisions to the manuscript to address all the questions and comments raised by the three reviewers. We highlights changes made to the original version by setting the text color to red. Our specific responses to each comment are as follows:

Responses to reviewers #1:

The manuscript entitled "Body Composition Patterns among Normal Glycemic, Pre-diabetic, Diabetic Health Chinese Adults in Community: NAHSIT 2013–2016" reports the association between body composition parameters and normal glycemic, prediabetic, and diabetic status in Chinese adult population sampled for the nationwide survey. Although the association between body composition parameters and each glycemic status was clearly described, the authors should discuss some additional points to emphasize a rationale of this study.

We are grateful for all of your constructive comments. In the revised manuscript, we addressed the all of the additional points in the following sections.

1. The authors should discuss the possible confounding factors of body composition and glycemic status. For example, smoking, alcohol and socioeconomic status could be confounding factors, especially in older adults (Osteoporosis and sarcopenia 2018;4:109-113, Lancet Diabetes Endocrinol 2015;3:958-967, Int J epidemiol 2011;40:804-818; J Diabetes Investig. 2020 Mar 29. doi: 10.1111/jdi.13260. etc).

Thank you for the comment. We agreed that smoking, alcohol, and socioeconomic status were possible confounding effects between body composition and glycemic status. We revised our manuscript by adding analyses for smoking, alcohol, and socioeconomic status.

For smoking, we categorized participants into current, ever, and no smokers. The ever and current smoker were more common in the diabetic groups but without statistical significance. (Please see Table 1, line 165-171, results section)

For alcohol, we obtained the average alcohol consumption (gram per day) and classified participants into non-drinker, <16, 16-35, 36-64, >64 g/day. Alcohol consumption between the diabetic and nondiabetic groups showed no significant difference. (Please see Table 1, line 165-171, results section) 

For socioeconomic status, we acquired z scores of personal income and education years. Individual socioeconomic status (iSES) was obtained by a sum of z scores of personal income and education years. Thereafter, the iSES was classified into the lowest, middle, and highest classes according to the tertiles of the z score. When compared to the diabetic group, the nondiabetic group had higher z score of iSES. (Please see and line 92-95, method section, and Table 1, line 165-171, results section)

Accordingly, the regression analyses for body composition markers that related to blood glucose (Table 3) and HbA1c (Table 5) were adjusted for age, sex, systolic and diastolic blood pressure, triglycerides, HDL, and iSES level. (Please see line 199, Table 3, and line 217, Table 5, results section)

2. The authors should consider the diabetic complications and anti-diabetes drugs to appropriately assess the relationship between body composition and glycemic status in patients with diabetes. Diabetic neuropathy and use of SU and glinides may affect skeletal muscle mass (J Am Med Dir Assoc 2016;17:896-901, J Diabetes Complications 2017;31:1465-1473, etc).

Thank you for the comment. We added the proportion of using anti-diabetes drugs in the diabetic group. For them, 50.9% (109/214) of the diabetic participants took anti-diabetic drugs. (Please see line 169-170, results section)

On the other hand, the enrolled participants in the NAHSIT study were healthy subjects, or people who had chronic diseases, such as diabetes, but generally these participants were in good medication control and favorable functional status. Compared to hospital-based subjects, our participants with diabetes were in the early disease status. Most of these healthy participants has not developed the diabetic complication yet. Diabetic complications were categorized into marco-vascular and micro-vascular complications. Among them, diabetic neuropathy was one of the micro-vascular complication. In most condition, type 1 and type 2 diabetes need > 20 or > 7 years of clinical course to develop the microvascular complications.[1] Since the NAHSIT was a community-based study and focused on healthy subjects, data of diabetic complication such diabetic neuropathy was not collected in this survey. We addressed this limitation in the discussion section. (Please see line 324-326, discussion section) 

3. The authors should discuss not only muscle mass but also muscle strength as well as exercise capacity. Dynapenia could also influence glycemic status (J epidemiol 2015 ; 25 : 656-662); and the authors should discuss the possible relationship in the manuscript.

Thank you for the comment. Since muscle strength was not measured and the data was unavailable in the NAHSIT 2013-2016, we addressed this limitation in the discussion section. (Please see discussion section, line 322-323)

4. The authors should discuss possible effects of ethnic differences on body composition and glycemic status. In the previous study, it was suggested that ethnic differences may affect body composition, diet and physical activity (Womens Health(Lond) 2015;11:913-27).

Thank you for the comment. We reviewed this literature and agreed the possible effects of ethnic difference (such as White, Black, and Chinese) on glycemic status and body composition. In fact, numerous countries or regions in east Asia were of a single ethnic group, such as China (91.6% of Han Chinese), South Korea (96% of Koreans), Japan (98% of Japanese), and Taiwan (95.4% of Han Chinese).

In fact, more than 95% of residents were in Taiwan were Han Chinese, and only 2.3% were Taiwanese aborigines. Therefore, the participants characteristic of our NAHSIT study was mono-ethnicity.[2] In our revised manuscript, we added the description of mono-ethnic characteristic in Taiwan in our discussion section. (Please see discussion section, line 328-333)

5. Since the phenotype of type 2 diabetes differ from those of other ethnic groups (Lancet Diabetes Endocrinol. 2016 Jan;4(1):2-3; Curr Diab Rep. 2015 Jun;15(6):602; J Diabetes Investig. 2016 Apr;7 Suppl 1(Suppl 1):102-9 etc). The authors should discuss possible etnic differences in their observed results.

Thank you for the comment. After reviewing these literatures, we agreed that the phenotype of type 2 diabetes or β-cell function and insulin sensitivity may vary by different ethnic groups, such as ethnic Caucasians, Japanese, Koreans, and Chinese. In Taiwan, 95.4% of residents were in fact Han Chinese, and 2.3% were Taiwanese aborigines (such as Ami, Atayal).[3] (Please see discussion section, line 300-305)

A subethnic study in Taiwan for β-cell function and insulin sensitivity also found no significant difference in the prevalence of diabetes and impaired glucose tolerance (IGT) between Aborigines and Chinese.[4] (Please see discussion section, line 328-333) 

Responses to reviewers #2:

Major:

1. In line 214-215, you are writing “lean mass…”. Therefore, not only comparison between glucose profile and body composition but also the comparison with an insulin level and HOMA-IR and the body composition is necessary.

We are grateful for all of your constructive comments. Since insulin level was not available in the NAHSIT 2013-2016 survey, we addressed this limitation. (Please see line 305-307, in the discussion section) 

On the other hand, we used the triglyceride glucose-waist circumference (TyG-WC) index as a surrogate marker for insulin resistance[5, 6]. According to a recent large scale nutritional survey for Asian population, the TyG index and TyG-related markers were highly correlated with homeostasis model assessment of insulin resistance (HOMA-IR).[6]

Of these TyG-related markers, we noticed that TyG-WC index had the highest diagnostic consistency to HOMA-IR for both male and female Asian participants. Therefore, we decided to conduct a sensitivity analyses for the association between body composition and insulin resistance with TyG-WC. A cut-off value of for TyG-WC ≥ 850 was defined as insulin resistance.

The sensitivity analysis of the association between body composition markers and TyG-WC was shown in Table S4 and S5. With increasing values of TyG-WC index, the participants showed decrease of lean mass weight and increase of fat mass.

(Please see line 150-159, methods section; line 226-231 results section; line 314-315, discussion section; line 321=326, discussion section; and Table S4-S5)

2. In line 243-244, you are writing “low lean mass percentage in abdomen…”. Results of RRR show that trunk lean and fat mass have a positive association with a glucose profile. Trunk in fat and lean has a positive association with glucose profile, and limb in fat and lean seems to have a negative association with glucose profile. Therefore, the discussion about this difference seems to be necessary.

Thank you for the great comment. We addressed (1) the different functional characteristics fat tissue between upper body (android) and lower body, and (2) the paradoxically positive association of trunk lean mass with FG and HbA1c levels in the discussion section. 

First, we considered these findings were compatible to previous studies indicating functional differences of fat tissue between upper body (android area) and lower body.[7-9] Previous studies found the opposite association of upper and lower body fat with 2-hour post-load glucose level of oral glucose tolerance test[9] and diabetic risk[10, 11]. Recent studies indicated this difference was medicated by different adipokines[8] and site-specific sets of genes between upper and low body fat tissues[7, 12], respectively

Second, the RRR also showed android lean mass was positively associated with FG and HbA1c. We found some studies showed the similar findings in their original data: diabetic participants with significantly higher trunk lean mass in comparison to non-diabetes.[11, 13] This should be cautious interpreted since DXA may severely overestimate trunk and android lean mass.[14, 15] When compared to magnetic resonance imaging (MRI) modality, a study found DXA overestimate 7% and 48% of trunk and android lean mass, respectively.[15] The association of trunk and android lean mass with FG and HbA1c should be reserved to answer with MRI. (Please see line 292-307, discussion section) 

Minor: 

1. In line 85, you are writing “…assessment of the nutritional status”. What kind of item did you use for nutritional evaluation, blood nutritional items (albumin, cholinesterase etc.) or nutrition score (MNA, CONUT etc.)?

Thank you for the comment. Continued from our previous 2005-2008 Nutrition and Health Survey in Taiwan (NAHSIT)[16], the 2013-2016 NAHSIT assessed both the (1) behavior and (2) health outcome indicators. 

For behavior indicators, we collected information of socioeconomic items, diet behavior and belief (including dietary recall, food frequency and habits, dietary and nutritional knowledge). 

For health outcome indicators, we measured the body composition by dual-energy X-ray absorptiometry and clinical biochemistry, including items of serum cholesterol (including total cholesterol, LDL-C and HDL-C), triglycerides, blood glucose, uric acid, CRP, creatinine, liver function tests, amylase, complete blood count, vitamins, minerals, serum pH, pCO2 , pO2, iron and ferritin, TIBC, phospholipid, homocysteine, BUN, alkaline phosphatase, PTH, and DNA from white blood cells for genetic analysis. (Please see line 85-90 in the revised manuscript)

2. In the section of Data Collection of Body Composition (117-125), the calculating formula is (body composition/1000)/weight *100, isn't it?

Thank you for the comment. We had revised the calculating formula. (Please see line 122-136)

3. In line 150, is the classification of the FG levels ≥126mg/dL, 100-126 mg/dL and <100mg/dL? It is necessary to conform them to data of Table2.

Thank you for the comment. The classification of FG levels were of ≥126 mg/dL, 100–125 mg/dL, and <100 mg/dL, which was conform to data of Table 2 in the revised manuscript. (Please see line186, and line 196, Table 2, results section)

4. In line 161, did you mistake “≥6.5%” for “>6.5%”?

Thank you for the comment. Th participants were categorized into three groups of ≥6.5%, 5.7–6.4%, and <5.7%. (Please see line 207, results section)

5. In line 205, you wrote "…increase trunk fat mass". But not only "trunk fat mass" but also "trunk lean mass" increased.

Thank you for the comment. We revised the text with adding “trunk lean mass.” (Please see line 270-271, discussion section)

6. Explanation of total region fat, total tissue fat, android (in fat) and gynoid (in fat) is necessary in the method.

Thank you for the comment. These terms were defined as the following.

• Total region fat (%): (Fat mass in limbs and trunk/ Body weight) × 100

• Total tissue fat (%): (Total fat mass/ Body weight) × 100

• Android in fat (%): (Android fat mass/Total fat mass) × 100

• Gynoid in fat (%): (Gynoid fat mass/Total fat mass) × 100

(Please see our revised manuscript, line 133-136, method section)

7. In Table 1, I think the data of Male is "number and ratio" not "mean and SD".

Thank you for the comment. The unit for data of male was revised. (Please see our revised manuscript, Table 1, results section)

8. The unit of weight is “kg” not “cm” (Table2-5).

Thank you for the comment. The unit of weight was corrected now. (Please see the revised manuscript, Table 2 to 5, results section)

9. Please revise the data of SD from 10.41 to 10.4 in the item of the waist in Table 2.

Thank you for the comment. We revised the data of SD from 10.41 to 10.4 in the item of the waist in Table 2. (Please see Table 2, results section)

10. Please revise “Statistical significance as P<0.5” to “Statistical significance as P<0.05” in Table 2.

Thank you for the comment. We made a revision as P <0.05 in Table 2. (Please see Table 2, results section)

11. I think that HbA1c is unnecessary for a title of Table3.

Thank you for the comment. We deleted HbA1c for a title of Table 3. (Please see Table 3, results section)

References

1. Association AD. 11. Microvascular Complications and Foot Care:. Diabetes Care. 2020;43(Suppl 1):S135-S51. doi: 10.2337/dc20-S011. PubMed PMID: 31862754.

2. https://en.wikipedia.org/wiki/Monoethnicity.

3. https://en.wikipedia.org/wiki/Demographics_of_Taiwan.

4. Chen HD, Shaw CK, Tseng WP, Chen HI, Lee ML. Prevalence of diabetes mellitus and impaired glucose tolerance in Aborigines and Chinese in eastern Taiwan. Diabetes Res Clin Pract. 1997;38(3):199-205. doi: 10.1016/s0168-8227(97)00104-6. PubMed PMID: 9483387.

5. Zheng S, Shi S, Ren X, Han T, Li Y, Chen Y, et al. Triglyceride glucose-waist circumference, a novel and effective predictor of diabetes in first-degree relatives of type 2 diabetes patients: cross-sectional and prospective cohort study. J Transl Med. 2016;14:260. Epub 2016/09/07. doi: 10.1186/s12967-016-1020-8. PubMed PMID: 27604550; PubMed Central PMCID: PMCPMC5015232.

6. Lim J, Kim J, Koo SH, Kwon GC. Comparison of triglyceride glucose index, and related parameters to predict insulin resistance in Korean adults: An analysis of the 2007-2010 Korean National Health and Nutrition Examination Survey. PLoS One. 2019;14(3):e0212963. Epub 2019/03/07. doi: 10.1371/journal.pone.0212963. PubMed PMID: 30845237; PubMed Central PMCID: PMCPMC6405083.

7. Karpe F, Pinnick KE. Biology of upper-body and lower-body adipose tissue--link to whole-body phenotypes. Nat Rev Endocrinol. 2015;11(2):90-100. Epub 2014/11/04. doi: 10.1038/nrendo.2014.185. PubMed PMID: 25365922.

8. Wu H, Qi Q, Yu Z, Sun Q, Wang J, Franco OH, et al. Independent and opposite associations of trunk and leg fat depots with adipokines, inflammatory markers, and metabolic syndrome in middle-aged and older Chinese men and women. J Clin Endocrinol Metab. 2010;95(9):4389-98. Epub 2010/06/02. doi: 10.1210/jc.2010-0181. PubMed PMID: 20519350.

9. Snijder MB, Dekker JM, Visser M, Bouter LM, Stehouwer CD, Yudkin JS, et al. Trunk fat and leg fat have independent and opposite associations with fasting and postload glucose levels: the Hoorn study. Diabetes Care. 2004;27(2):372-7. doi: 10.2337/diacare.27.2.372. PubMed PMID: 14747216.

10. Tatsukawa Y, Misumi M, Kim YM, Yamada M, Ohishi W, Fujiwara S, et al. Body composition and development of diabetes: a 15-year follow-up study in a Japanese population. Eur J Clin Nutr. 2018;72(3):374-80. Epub 2018/01/23. doi: 10.1038/s41430-017-0077-7. PubMed PMID: 29362458.

11. Lee JS, Auyeung TW, Leung J, Kwok T, Leung PC, Woo J. The effect of diabetes mellitus on age-associated lean mass loss in 3153 older adults. Diabet Med. 2010;27(12):1366-71. doi: 10.1111/j.1464-5491.2010.03118.x. PubMed PMID: 21059088; PubMed Central PMCID: PMCPMC3059762.

12. Pinnick KE, Nicholson G, Manolopoulos KN, McQuaid SE, Valet P, Frayn KN, et al. Distinct developmental profile of lower-body adipose tissue defines resistance against obesity-associated metabolic complications. Diabetes. 2014;63(11):3785-97. Epub 2014/06/19. doi: 10.2337/db14-0385. PubMed PMID: 24947352.

13. Kim KS, Park KS, Kim MJ, Kim SK, Cho YW, Park SW. Type 2 diabetes is associated with low muscle mass in older adults. Geriatr Gerontol Int. 2014;14 Suppl 1:115-21. doi: 10.1111/ggi.12189. PubMed PMID: 24450569.

14. Buckinx F, Landi F, Cesari M, Fielding RA, Visser M, Engelke K, et al. Pitfalls in the measurement of muscle mass: a need for a reference standard. J Cachexia Sarcopenia Muscle. 2018;9(2):269-78. Epub 2018/01/19. doi: 10.1002/jcsm.12268. PubMed PMID: 29349935; PubMed Central PMCID: PMCPMC5879987.

15. Rankin KC, O'Brien LC, Gorgey AS. Quantification of trunk and android lean mass using dual energy x-ray absorptiometry compared to magnetic resonance imaging after spinal cord injury. J Spinal Cord Med. 2019;42(4):508-16. Epub 2018/02/20. doi: 10.1080/10790268.2018.1438879. PubMed PMID: 29461936; PubMed Central PMCID: PMCPMC6718191.

16. Tu SH, Chen C, Hsieh YT, Chang HY, Yeh CJ, Lin YC, et al. Design and sample characteristics of the 2005-2008 Nutrition and Health Survey in Taiwan. Asia Pac J Clin Nutr. 2011;20(2):225-37. PubMed PMID: 21669592.

---

## [Decision Letter · Decision Letter 1]

9 Oct 2020

Body Composition Patterns among Normal Glycemic, Pre-diabetic, Diabetic Health Chinese Adults in Community: NAHSIT 2013–2016

PONE-D-20-13486R1

Dear Dr. Bai,

We’re pleased to inform you that your manuscript has been judged scientifically suitable for publication and will be formally accepted for publication once it meets all outstanding technical requirements.

Kind regards,

Cheng Hu

Academic Editor

PLOS ONE

Additional Editor Comments (optional):

Reviewers' comments:

Reviewer's Responses to Questions

**Comments to the Author**

1. If the authors have adequately addressed your comments raised in a previous round of review and you feel that this manuscript is now acceptable for publication, you may indicate that here to bypass the “Comments to the Author” section, enter your conflict of interest statement in the “Confidential to Editor” section, and submit your "Accept" recommendation.

Reviewer #1: All comments have been addressed

Reviewer #2: All comments have been addressed

2. Is the manuscript technically sound, and do the data support the conclusions?

Reviewer #1: Yes

Reviewer #2: Yes

3. Has the statistical analysis been performed appropriately and rigorously? 

Reviewer #1: Yes

Reviewer #2: Yes

4. Have the authors made all data underlying the findings in their manuscript fully available?

Reviewer #1: Yes

Reviewer #2: Yes

5. Is the manuscript presented in an intelligible fashion and written in standard English?

Reviewer #1: Yes

Reviewer #2: Yes

6. Review Comments to the Author

Reviewer #1: (No Response)

Reviewer #2: Thank you for revision and comment. I confirmed contents again.

There are no additional question and revision.

This paper is acceptable.

7. PLOS authors have the option to publish the peer review history of their article (what does this mean?). If published, this will include your full peer review and any attached files.

Reviewer #1: No

Reviewer #2: No

---

## [Editor Report · Acceptance letter]

22 Oct 2020

PONE-D-20-13486R1 

Body composition patterns among normal glycemic, pre-diabetic, diabetic health Chinese adults in community: NAHSIT 2013–2016 

Dear Dr. Bai:

I'm pleased to inform you that your manuscript has been deemed suitable for publication in PLOS ONE. Congratulations! Your manuscript is now with our production department. 

Kind regards, 

on behalf of

Dr. Cheng Hu 

Academic Editor

PLOS ONE